# Segmentation of time series in up- and down-trends using the epsilon-tau procedure with application to USD/JPY foreign exchange market data

Arthur Matsuo Yamashita Rios de Sousa[1], Hideki Takayasu[1,2], Misako Takayasu[1,3]*

**1** Institute of Innovative Research, Tokyo Institute of Technology, Yokohama, Kanagawa, Japan, **2** Sony Computer Science Laboratories, Tokyo, Japan, **3** Department of Mathematical and Computing Science, School of Computing, Tokyo Institute of Technology, Yokohama, Kanagawa, Japan

* takayasu.m.aa@m.titech.ac.jp

## Abstract

We propose the epsilon-tau procedure to determine up- and down-trends in a time series, working as a tool for its segmentation. The method denomination reflects the use of a tolerance level $\varepsilon$ for the series values and a patience level $\tau$ in the time axis to delimit the trends. We first illustrate the procedure in discrete random walks, deriving the exact probability distributions of trend lengths and trend amplitudes, and then apply it to segment and analyze the trends of U.S. dollar (USD)/Japanese yen (JPY) market time series from 2015 to 2018. Besides studying the statistics of trend lengths and amplitudes, we investigate the internal structure of the trends by grouping trends with similar shapes and selecting clusters of shapes that rarely occur in the randomized data. Particularly, we identify a set of down-trends presenting similar sharp appreciation of the yen that are associated with exceptional events such as the Brexit Referendum in 2016.

## Introduction

Time series segmentation consists in dividing the original time series in segments with similar behavior according to some criteria and it can be either a preprocessing step in order to represent the time series more efficiently or a data mining technique on its own, able to extract information about the dynamics of the underlying phenomenon [1, 2]. Segmentation methods have been utilized to analyze time series of diverse backgrounds, including biological, climate, remote sensing and crime-related data [3–9].

Especially in the context of finance, various techniques to segment time series by identifying periods with similar behavior or by finding switching points were developed [10–17]. A particularly relevant category of such procedures is the segmentation of time series in up- and down trends, since the identification of periods presenting general tendency of increase or decrease is fundamental for risk management and to recognize investment opportunities. Often referred as drawdowns and drawups, there are several methods to determine trends in

**Data Availability Statement:** The raw data used in this study was purchased from the EBS Service

Company Limited, with no special access privileges. Due to the contract between EBS and us, the authors are not allowed to distribute the raw data. Following the same procedure as the authors, those researchers interested in analyzing similar data sets are recommended to contact the EBS Service Company Limited about the availability and purchase of the data (see https://www.cmegroup.com/tools-information/contacts-list/ebs-support.html).

**Funding:** This study was supported by the Joint Collaborative Research Laboratory for MUFG AI Financial Market Analysis. The funder had no role in study design, data collection and analysis, decision to publish, or preparation of the manuscript. Author HT is employed by Sony Computer Science Laboratories, Inc, which provided support in the form of salaries for author HT, but did not have any additional role in the study design, data collection and analysis, decision to publish, or preparation of manuscript. The specific roles of this author are articulated in the 'author contributions' section.

**Competing interests:** The authors of this manuscript have the following competing interests: MT receives research funding (the Joint Collaborative Research Laboratory for MUFG AI Financial Market Analysis) from a commercial source (Mitsubishi UFJ financial group) as consultancy of AI trading strategies; HT is employed by Sony Computer Science Laboratories, Inc. There are no patents, products in development or marketed products to declare. This does not alter our adherence to all the PLOS ONE policies on sharing data and materials, as detailed online in the guide for authors.

financial time series [18–22]. A strict definition of drawdown (drawnup) is the continuous decrease (increase) of the series values, terminated by any movement in the opposite direction. Such definition, however, may not be adequate to correctly assess market risks because it is too sensitive to noise. Addressing this limitation, the epsilon-drawdown method was proposed; its improvement is based on the introduction of a tolerance $\varepsilon$ within which fluctuations are ignored and the trend is not interrupted [23–25]. Another way to reduce the noise sensitivity is to ignore movements within time horizon $\tau$, method suggested but not discussed in [23]. In this work, we unify and extend those previous ideas and propose the epsilon-tau procedure, that simultaneously makes use of a tolerance level $\varepsilon$ and a patience level $\tau$ to determine up- and down-trends in time series.

As for the paper outline, in the following section we present the epsilon-tau procedure as a method to define the up- or down-trend associated with a given reference point in a time series. We then illustrate its application in simple random walks. We derive exact expression for the marginal probability distributions of trend lengths and trend amplitudes and explain how to employ the epsilon-tau procedure to segment time series. Finally, we apply the segmentation to analyze foreign exchange data consisting of U.S. dollar/Japanese yen market time series from 2015 to 2018. We pay special attention to the internal structure of the trends, performing a systematic investigation of trend shapes that occur in the market time series and introducing an approach based on the Fisher's exact test to select abnormal shapes that are rarely produced when the data is randomized.

## Epsilon-tau procedure

Consider a time series $\{x_{t \in \mathbb{Z}}\}$ and a reference point $t = m$ with value $x_m$. We say that the trend associated with the reference point is an up-trend if $x_{m+1} - x_m > 0$ and it is a down-trend if $x_{m+1} - x_m < 0$; if $x_{m+1} - x_m = 0$, the trend is not determined for that reference point.

For the up-trend case (analogous for the down-trend case), the epsilon-tau procedure with tolerance level $\varepsilon > 0$ and patience level $\tau \geq 1$—both possibly time-dependent—consists in comparing values $x_t$, $t \geq m + 1$, with the previous rightmost maximum value $\max_{m + 1 \leq t' \leq t}\{x_{t'}\}$. The procedure stops at $t = t^*$ when one of the following conditions is met:

(a) value of time series reaches tolerance level $\varepsilon$ (Fig 1a):

$$\max_{m+1 \leq t' \leq t^*}\{x_{t'}\} - x_{t^*} \geq \varepsilon; \tag{1}$$

(b) time between consecutive maximum values reaches patience level $\tau$ (Fig 1b):

$$t^* - arg \max_{m+1 \leq t' \leq t^*}\{x_{t'}\} \geq \tau. \tag{2}$$

The trend determined from this procedure is the up-trend $[m + 1, m + \ell]$ of length $\ell = arg\max_{m + 1 \leq t' \leq t^*}\{x_{t'}\} - m$, $\ell \geq 1$, and amplitude $a = x_{m + \ell} - x_m$, $a > 0$, where $x_{m + \ell} = \max_{m + 1 \leq t' \leq t^*}\{x_{t'}\}$. Observe that the trend ends in $m + \ell$ and not in $t^*$; the point $t^*$ indicates the stop of the procedure, with at least one point and at most $\tau$ points beyond the end of the trend needing to be checked in order to determine it.

We remark that the epsilon-tau procedure does not require a predetermined functional form by which trends are approximated, as it happens, for instance, in piecewise linear methods, where trends are approximated by straight lines [2]; this is an important feature that allow us to explore the diversity of possible trend shapes.

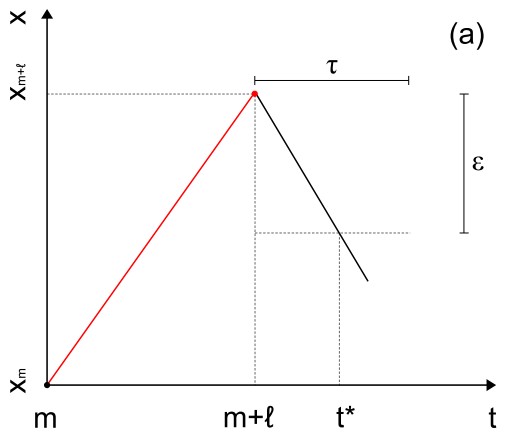 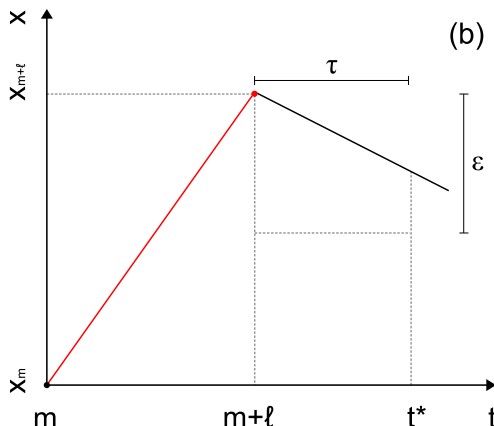

**Fig 1. Stop conditions of the epsilon-tau procedure for the up-trend case (analogous for the down-trend case).** Procedure stops when: (a) value of time series reaches tolerance level $\varepsilon$; or (b) time between consecutive maximum values reaches patience level $\tau$. It defines the up-trend (red) of length $\ell \geq 1$ and amplitude $a = x_{m+\ell} - x_m > 0$.

The epsilon-drawdown method used in [23–25] can be regarded as a particular instance of the epsilon-tau procedure with infinite patience level $\tau \to \infty$. Developed in a financial context, the cited works selected a time-dependent tolerance level $\varepsilon$ proportional to the volatility (measure of price variation over time) estimated over a preceding time window, being more permissive when the market presents high volatility and becoming stricter during calmer periods. Instead, we use throughout this paper the time-dependent tolerance level $\varepsilon = \max_{\{m+1 \leq t' \leq t\}} x_{t'} - x_m$ for the up-trend case (analogous for the down-trend case). Using such tolerance level, stop condition (a) is translated as $x_{t^*} \leq x_m$, i.e., the procedure stops if the reference value $x_m$ is reached, and then all points in the up-trend $[m+1, m+\ell]$ have values always in between the reference value $x_m$ and the maximum value $x_{m+\ell}$: $x_m < x_t \leq x_{m+\ell}, \forall t \in [m+1, m+\ell]$. Such choice of tolerance level is related to trading psychology by setting the initial price level as the tolerance to keep believing that an up- or down-trend will recover and continue (naturally, this tolerance can be set higher for more aggressive traders or lower for risk averse ones). As for the patience level $\tau$, we use it here as a time constant parameter and it is interpreted as the interval of time that the observer is willing to wait to confirm that a up- or down-trend has ended. The choice of its value is then connected to the characteristics of the observer and her/his intentions; for example, if the aim is the development of real-time applications, a small $\tau$ is more suitable, but for historical analysis, larger $\tau$ values can provide valuable information.

In the next section, we apply the epsilon-tau procedure with the described tolerance and patience level to simple random walks to illustrate the theoretical study of the trend length and trend amplitude probability distributions for different values of $\tau$ and to introduce the time series segmentation using the procedure.

## Up- and down-trends in random walks

Take the random walk:

$$x_t = x_{t-1} + \xi_t, \tag{3}$$

where the independent and identically distributed increments $\xi_t$ can take value $+1$ with probability $p$, $-1$ with probability $q$, or $0$ with probability $r = 1 - p - q$.

For the strict definition of drawdown, which corresponds to tolerance level $\varepsilon$ approaching zero or patience level $\tau = 1$ in the epsilon-tau procedure, it was shown in [22] that the trend length and trend amplitude marginal probability distributions are asymptotically exponential when the time series increments are independent with a non-heavy-tailed distribution. Such asymptotic exponential behavior also occurs when applying the epsilon-tau procedure to the considered random walk, as we show next by presenting the exact expressions for the probability distributions and numerical simulations. The detailed derivation of the distributions can be found in S1 Appendix.

## Trend length marginal probability distribution

The up-trend length $\ell$ probability distribution for patience level $\tau = 1$ is given by (analogous for down-trend):

$$P(up, \ell; \tau = 1) = p(p + r)^{\ell-1} q. \tag{4}$$

For patience level $\tau = 2$, the distribution is:

$$P(up, \ell; \quad \tau = 2) = pr^{\ell-1}q + \frac{p^2 q(r + q)}{\sqrt{(p + q)^2 + 4pq}}$$

$$\times \left\{ -\left( \frac{p + r - \sqrt{(p + q)^2 + 4pq}}{p - r - \sqrt{(p + q)^2 + 4pq}} \right) \left[ \left( \frac{p + r - \sqrt{(p + q)^2 + 4pq}}{2} \right)^{\ell-1} - r^{\ell-1} \right] \right.$$

$$\left. + \left( \frac{p + r + \sqrt{(p + q)^2 + 4pq}}{p - r + \sqrt{(p + q)^2 + 4pq}} \right) \left[ \left( \frac{p + r + \sqrt{(p + q)^2 + 4pq}}{2} \right)^{\ell-1} - r^{\ell-1} \right] \right\}. \tag{5}$$

For patience level $\tau \geq 3$, the computation becomes involved and we do not derive those distributions here. But in Fig 2 we show length distributions from numerical simulations for different values of $\tau$ and random walk parameters. We observe agreement with the theoretical distributions for cases $\tau = 1$ and $\tau = 2$ and the presence of exponential tails in all cases, where the value of $\tau$ controls the decay rate.

## Trend amplitude marginal probability distribution

The up-trend amplitude $a$ probability distribution for arbitrary patience level $\tau$ is given by the following expression:

$$P(up, a; \tau) = \left[ \prod_{k=0}^{a-1} \frac{p}{1 - \sum_{j=1}^{\tau} P(z_{jk})} \right] \left[ \sum_{j=1}^{\tau} P(s_{ja}^{(\varepsilon)}) + P(s_{\tau(a-1)}^{(\tau)}) \right], \tag{6}$$

where (using results on lattice path enumeration and powers of tridiagonal toeplitz matrices [26–28]):

$$P(z_{jk}) = \begin{cases} 0, & \text{if } j \geq 2, k = 0; \\ r, & \text{if } j = 1, k \geq 0; \\ \frac{2pq}{k+1} \sum_{u=1}^{k} \lambda_{\frac{u}{k+1}}^{j-2} \sin^2\left(\frac{u\pi}{k+1}\right), & \text{if } j \geq 2, k \geq 1, \end{cases} \tag{7}$$

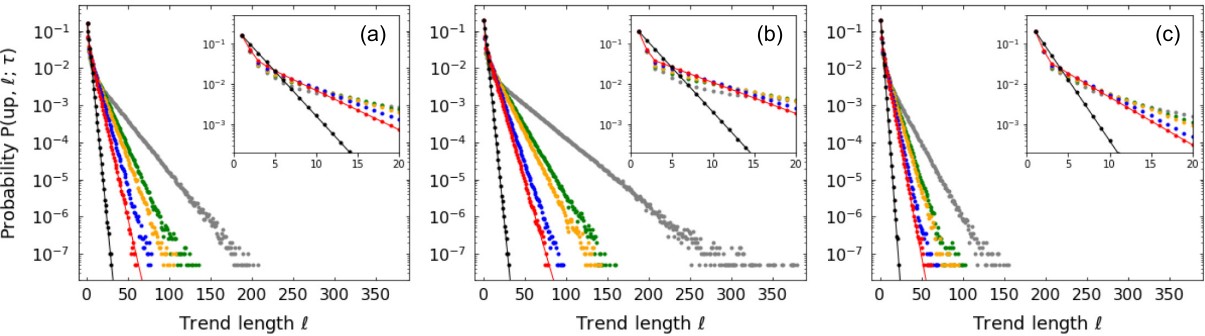

**Fig 2. Up-trend length $\ell$ probability distributions for random walks.** Distributions for patience levels $\tau = 1$ (black), $\tau = 2$ (red), $\tau = 3$ (blue), $\tau = 4$ (orange), $\tau = 5$ (green), $\tau = 10$ (gray) and for random walk parameters: (a) $p = 0.4$, $q = 0.4$; (b) $p = 0.5$, $q = 0.4$; (c) $p = 0.4$, $q = 0.5$. Symbols refer to results from numerical simulations and lines represent theoretical values. Insets detail distributions for small $\ell$.

with $\lambda_{\frac{u}{k+1}} = r + 2\sqrt{pq}\cos(\frac{u\pi}{k+1})$.

$$P(s_{jk}^{(\varepsilon)}) = \begin{cases} 0, & \text{if } j \geq 1, k = 0 \text{ or } j = 1, k \geq 2 \text{ or } j \geq 2, k = 1; \\ q, & \text{if } j = 1, k = 1; \\ \frac{2q^2}{k}\left(\frac{q}{p}\right)^{\frac{k-2}{2}}\sum_{u=1}^{k-1}\lambda_{\frac{u}{k}}^{j-2}\sin\left(\frac{u\pi}{k}\right)\sin\left(\frac{(k-1)u\pi}{k}\right), & \text{if } j \geq 2, k \geq 2. \end{cases} \tag{8}$$

And:

$$P(s_{jk}^{(\tau)}) = \begin{cases} 0, & \text{if } j \geq 1, k = 0; \\ \sum_{u=1}^{k}\frac{2q}{k+1}\left(\frac{q}{p}\right)^{\frac{u-1}{2}}\sum_{v=1}^{k}\lambda_{\frac{v}{k+1}}^{j-1}\sin\left(\frac{uv\pi}{k+1}\right)\sin\left(\frac{v\pi}{k+1}\right), & \text{if } j \geq 1, k \geq 1. \end{cases} \tag{9}$$

For large amplitudes $a > \tau$ we can highlight the exponential behavior and write:

$$P(up, a; \tau) = \left[\prod_{k=0}^{\tau-1}\frac{p}{1 - \sum_{j=1}^{\tau}P(z_{jk})}\right]\left[\frac{p}{1 - \sum_{j=1}^{\tau}P(z_{jk})}\right]^{a-\tau}P(s_{\tau\tau}^{(\tau)}). \tag{10}$$

Fig 3 shows amplitude distributions from numerical simulations for different values of $\tau$ and random walk parameters. Simulation results agree with theoretical distributions for all values of $\tau$, which also control the decay rate of the exponential tails.

## Time series segmentation

The epsilon-tau procedure can be straightforwardly employed to segment a time series in alternating up- and down-trends by setting the end of a trend as the reference point for the next one. An up-trend is always followed by a down-trend and vice-versa (except in the end of the time series, in which the last trend may not be determined due to the finite size of the series). An example of segmented random walk using patience level $\tau = 7200$ is displayed in Fig 4a, where up-trends are colored red and down-trends, blue. Fig 4b shows the dependence of the segmentation result on the patience level $\tau$, with larger values of $\tau$ producing a coarser segmentation with larger trends on average.

Note that the marginal probability distributions of length $\ell$ and amplitude $a$ of trends from the segmentation of a random walk time series differ from the ones derived previously. Such

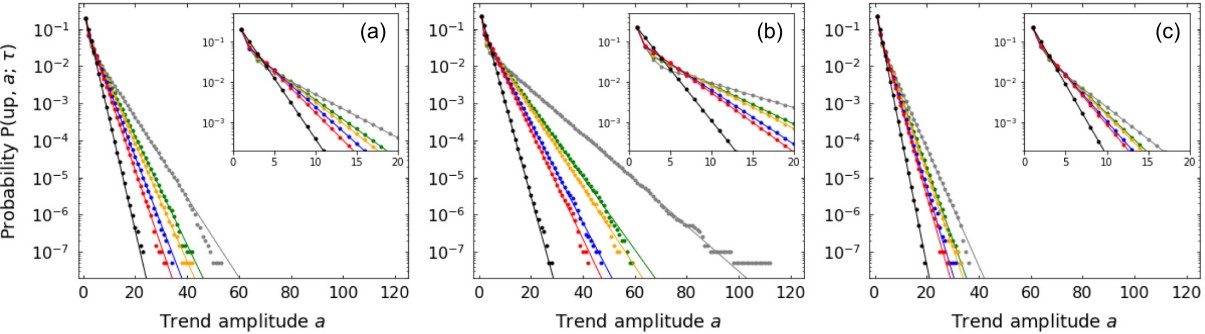

**Fig 3. Up-trend amplitude $a$ probability distributions for random walks.** Distributions for patience levels $\tau = 1$ (black), $\tau = 2$ (red), $\tau = 3$ (blue), $\tau = 4$ (orange), $\tau = 5$ (green), $\tau = 10$ (gray) and for random walk parameters: (a) $p = 0.4$, $q = 0.4$; (b) $p = 0.5$, $q = 0.4$; (c) $p = 0.4$, $q = 0.5$. Symbols refer to results from numerical simulations and lines represent theoretical values. Insets detail distributions for small $a$.

difference arises from the fact that the reference point used to define a given trend is not arbitrary anymore, but it is conditioned to be the end of the previous trend. Figs 5 and 6 make explicit the distinction between the two cases for trend length and trend amplitude, respectively: gray symbols correspond to trends produced by taking arbitrary reference points and black symbols indicate trends resulting from the time series segmentation. In the segmentation case, the stop conditions of the epsilon-tau procedure acting in a trend restricts the next trend, strongly affecting the probability of the small ones (both in length and in amplitude);

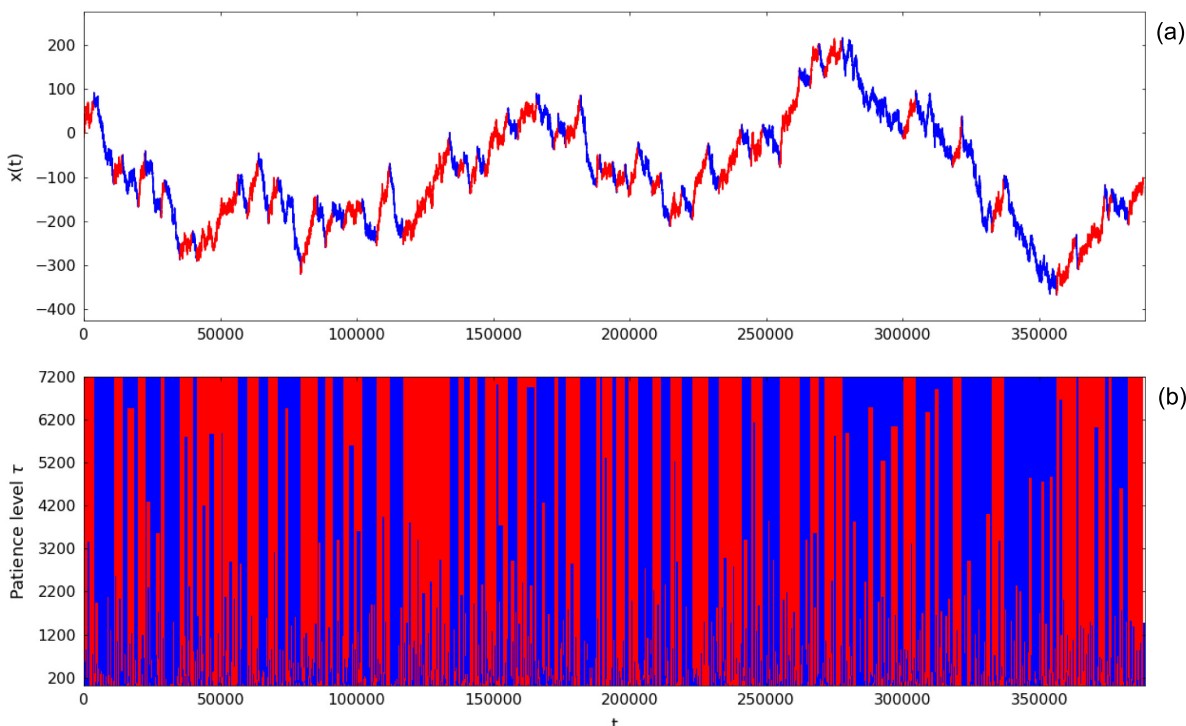

**Fig 4. Time series segmentation of a random walk realization.** (a) Up- and down-trends segmentation using patience level $\tau = 7200$ for random walk parameters $p = 0.4$, $q = 0.4$. (b) Segmentation results for different patience levels $\tau$. Red indicates up-trends, blue indicates down-trends and light-gray (in (a)) or white (in (b)) shows points where the trend is not determined (in the end of the time series—an effect of the finite size of the series).

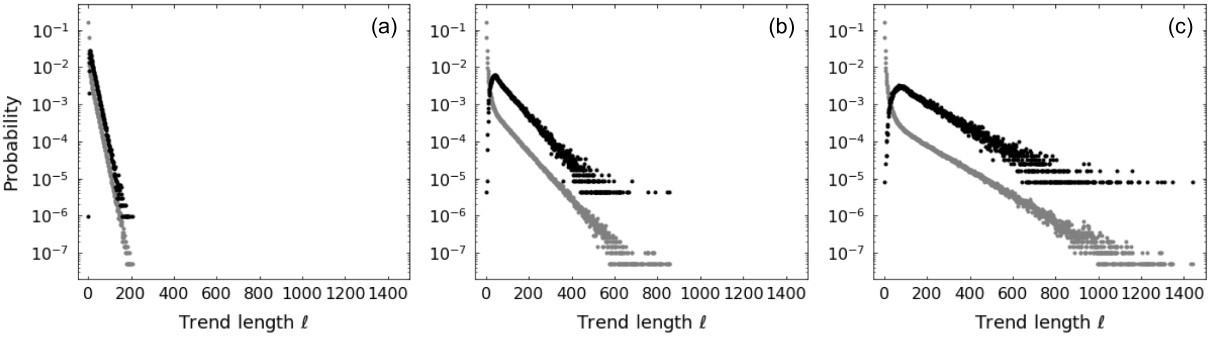

**Fig 5. Comparison between up-trend length $\ell$ probability distributions for random walk with parameters $p = 0.4$, $q = 0.4$.** Distributions considering arbitrary reference point (gray) and considering the trends obtained from time series segmentation (black) using patience levels: (a) $\tau = 10$; (b) $\tau = 50$; and (c) $\tau = 100$. Symbols refer to results from numerical simulations.

nevertheless, the decay rates of the exponential tails appear to be the same as the arbitrary reference point case.

## Up- and down-trends in financial time series

We now use the time series segmentation by the epsilon-tau procedure to analyze actual financial time series from the foreign exchange market, which has the largest trading volume among all financial markets (6.6 trillion U.S. dollar per day as reported in April 2019 [29]). We use the dataset from the Electronic Broking Service (EBS), one of the main trading platforms in this market, continuously open during weekdays from Sunday 21:00:00 GMT to Friday 21:00:00 GMT. Traders in this platform, mostly banks and financial institutions, can place buy and sell quotes for a given currencies pair; the mid-quote is defined at each time as the average of the highest buy quote and the lowest sell quote and a deal occurs when there is a match between those quotes. We study here the mid-quote time series of the currency pair U.S. dollar (USD) and Japanese yen (JPY) in a time resolution of one second from 2015 to 2018: 51 weeks of 2015, from 2015 January 05 to 2015 December 25; 52 weeks of 2016, from 2016 January 04 to 2016 December 30; 52 weeks of 2017, from 2017 January 02 to 2017 December 29; and 51 weeks of 2018, from 2018 January 08 to 2018 December 28 (each week from Monday 00:00:00 GMT to Friday 12:00:00 GMT).

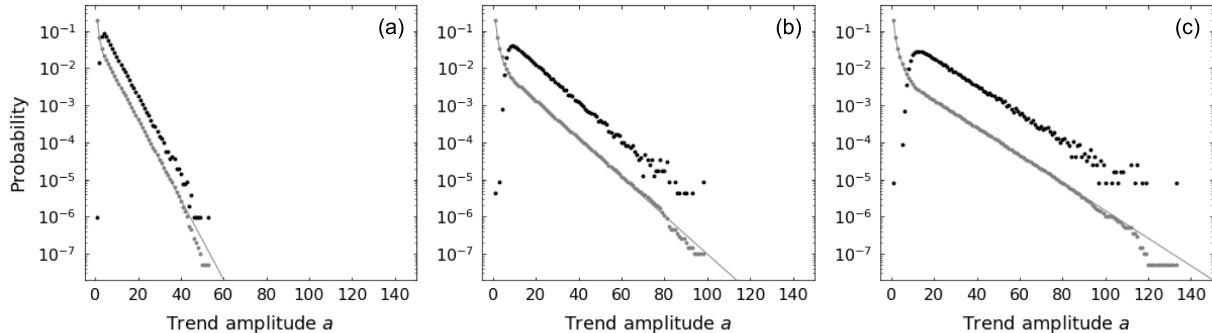

**Fig 6. Comparison between up-trend amplitude $a$ probability distributions for random walk with parameters $p = 0.4$, $q = 0.4$.** Distributions considering arbitrary reference point (gray) and considering the trends obtained from time series segmentation (black) using patience levels: (a) $\tau = 10$; (b) $\tau = 50$; and (c) $\tau = 100$. Symbols refer to results from numerical simulations and lines represent theoretical values.

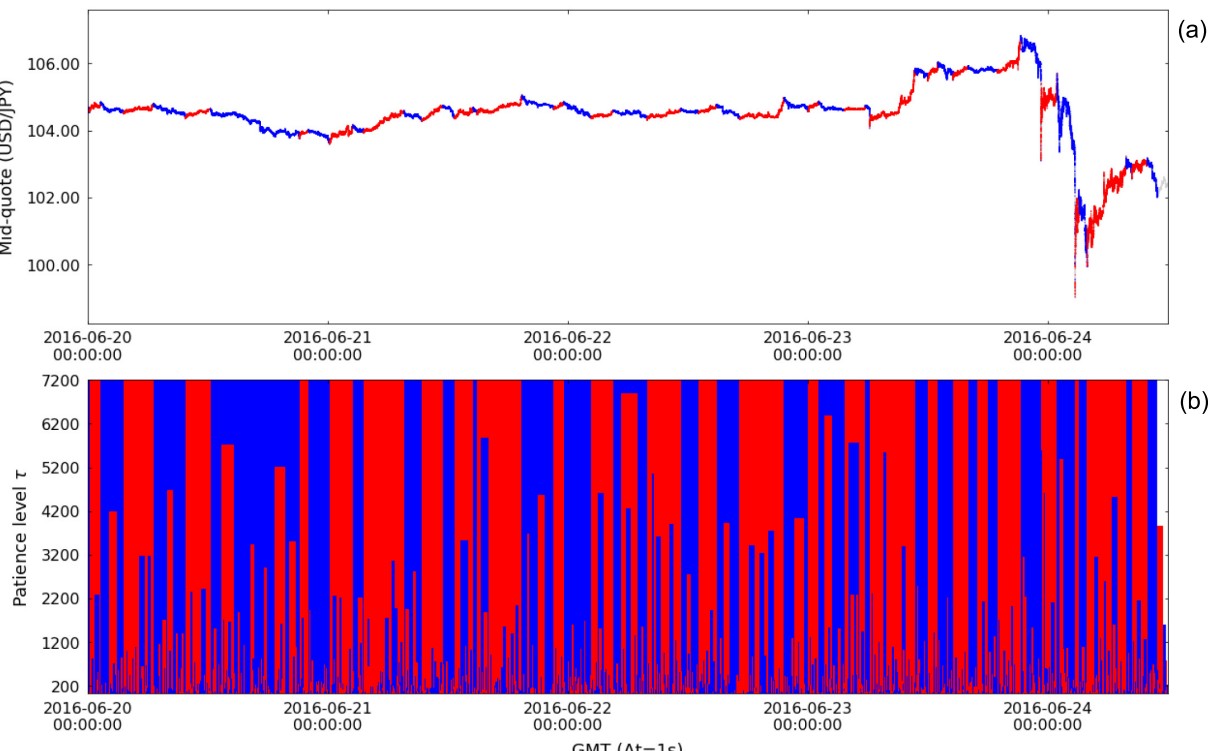

**Fig 7. Time series segmentation of the mid-quote time series of the currency pair USD/JPY during the week from June 20 2016 00:00:00 GMT to June 24 2016 12:00:00 GMT, when the Brexit Referendum took place.** (a) Up- and down-trends segmentation using patience level $\tau$ = 7200 (2h). (b) Segmentation results for different patience levels $\tau$. Red indicates up-trends, blue indicates down-trends and light-gray (in (a)) or white (in (b)) shows points where the trend is not determined.

The considered period includes several events that impacted the financial markets, the Brexit Referendum in June 2016 being one among the most relevant [30, 31]. In the foreign exchange market, this event caused the pound sterling to fall against the U.S. dollar to its lowest level since 1985 and a strong appreciation of the Japanese yen. We use the week when the Brexit Referendum took place to illustrate the segmentation of financial time series. Fig 7a displays the segmentation of the mid-quote time series of the currency pair USD/JPY in the referred week, in which the surge of the Japanese yen against the U.S. dollar reflects the market realization of the decision of the United Kingdom to leave the European Union in the night of June 23 and morning of June 24. In this example of financial time series segmentation, we use patience level $\tau$ = 7200 (2h) for better visualization of the trends in the one week time frame; Fig 7b shows the segmentation results for different values of $\tau$. Focusing on the yen surge, Fig 8 details the effect of the value of $\tau$ on the up- and down-trends of the segmented time series, with small $\tau$ emphasizing the microtrends and large $\tau$, the trends regarded as macrotrends (for the one week time frame).

## Trend length and trend amplitude marginal cumulative probability distributions

We start the statistical analysis of the trends obtained from the segmentation of the financial time series for the whole four years period by constructing the marginal (complementary) cumulative distributions of trend lengths $\ell$ and absolute trend amplitudes $|a|$ for three values of patience level: $\tau$ = 60 (1min) (highlighting microtrends), $\tau$ = 600 (10min) (intermediate

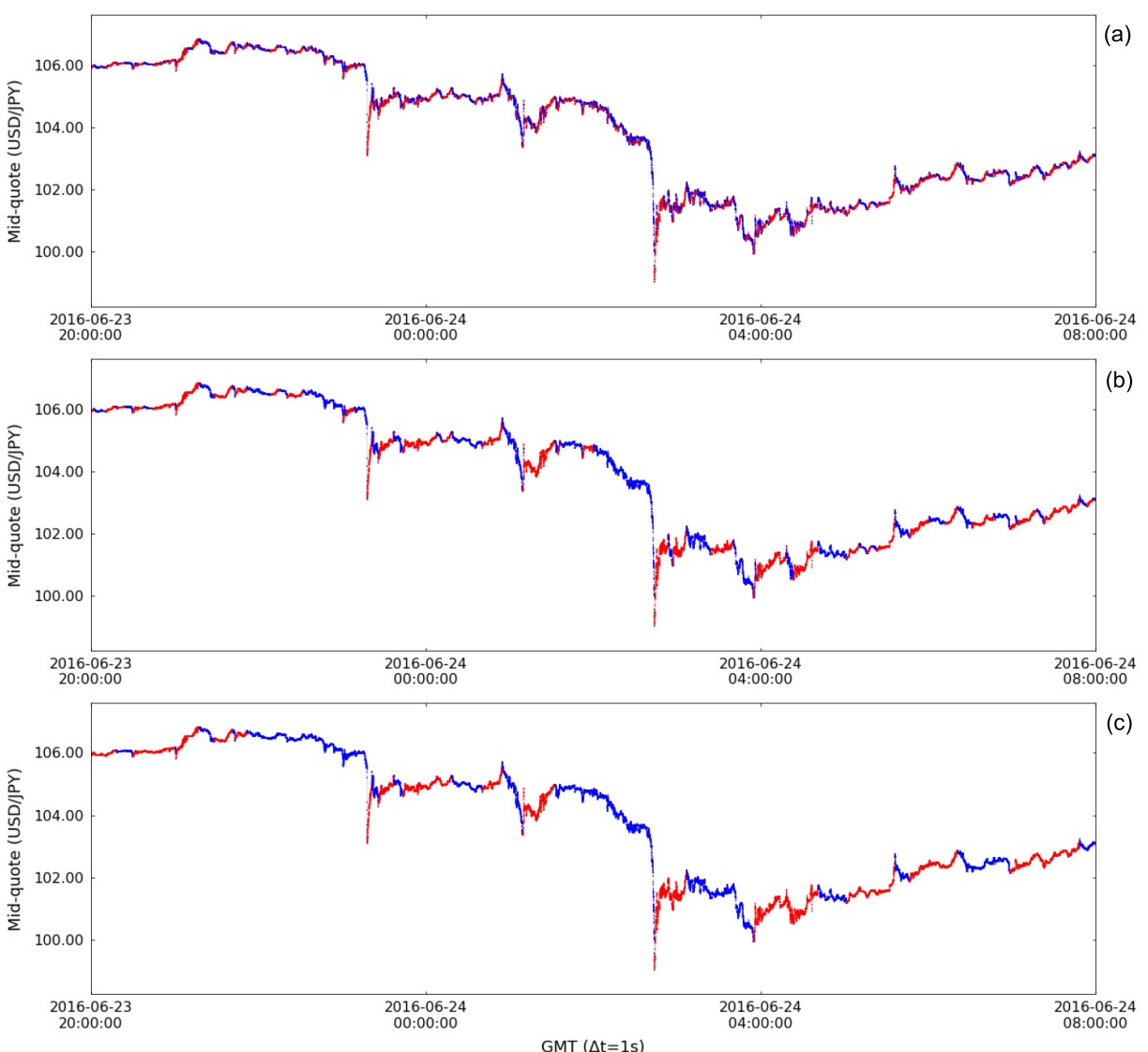

**Fig 8. Time series segmentation of the mid-quote time series of the currency pair USD/JPY during the 2016 Brexit Referendum.**
Segmentation results depend on the used patience level: (a) $\tau$ = 60 (1min); (a) $\tau$ = 600 (10min); and (a) $\tau$ = 1800 (30min).

case) and $\tau$ = 1800 (30min) (highlighting macrotrends). We separate the up- and down-trend cases and also constructs the distributions for randomized data. For the randomization, we shuffle the increments of the mid-quote time series of each week individually; because of the high fraction of zero increments in the one second resolution mid-quote time series (82.87% of zero, 8.56% of positive and 8.57% of negative increments), we consider two kinds of randomization: fixed zeros randomization, where we fix the zero increments in their original positions and shuffle only the positive and negative ones, and total randomization, where all increments are shuffled.

The distributions of trend lengths $\ell$ are present in Fig 9. For the small value of $\tau$ = 60 (1min) (Fig 9a), the distributions corresponding to the totally randomization case decay exponentially while the one of the fixed zeros randomization are similar to the distributions of trends from the original mid-quote data, which have tails heavier than an exponential both for

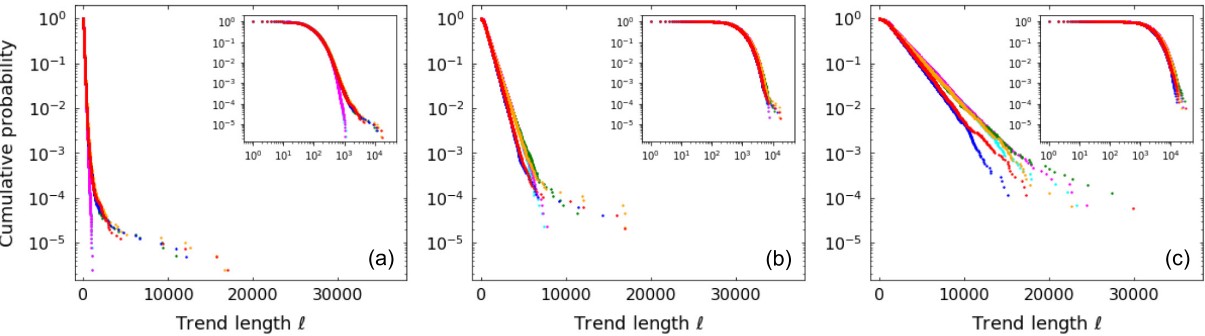

**Fig 9. Trend length $\ell$ cumulative probability distributions for mid-quote time series of the currency pair USD/JPY from 2015 to 2018.** Distributions for up-trends (red) and down-trends (blue) obtained from the segmentation of the mid-quote data, for up-trends (orange) and down-trends (green) obtained from the segmentation of the randomized mid-quote data with fixed zeros, and for up-trends (magenta) and down-trends (cyan) obtained from the segmentation of the totally randomized mid-quote data using patience levels: (a) $\tau = 60$ (1min); (a) $\tau = 600$ (10min); and (a) $\tau = 1800$ (30min). Insets show log-log plots.

up- and down-trends. The similarity between the fixed zeros randomization case and the original data indicates that sequences of zero increments control the length of microtrends; in fact, the trend length distributions for small $\tau$ shed light on the silent periods of the market, i.e., when there is no trading activity that changes the mid-quote. The effect of the sequences of zeros is reduced for large values of $\tau$ (see results for $\tau = 1800$ (30min) Fig 9b), for which both cases of randomization yield similar distributions with exponential tails and the distributions corresponding to the original data lose the heavy tail, presenting an approximate exponential decay but distinct from the random cases. We also note that the probabilities of long up- and down-trends significantly differ from each other, with long up-trends occurring more frequently than long down-trends.

Fig 10 shows the distributions of absolute trend amplitudes $|a|$, not presenting major qualitative differences for different values of $\tau$. The distributions for both randomization types decays exponentially, confirming that the sequences of zeros increments are less important for the amplitudes. The distributions corresponding to the original market data decay slower than the exponential ones of the random cases, with tails approximated by power-laws, which is in accordance with the results using the epsilon-drawdown method in financial markets [24].

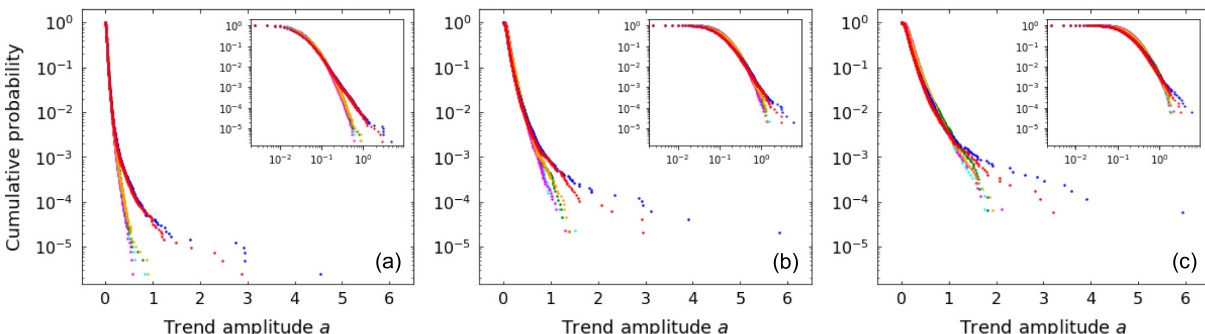

**Fig 10. Absolute trend amplitude $|a|$ cumulative probability distributions for mid-quote time series of the currency pair USD/JPY from 2015 to 2018.** Distributions for up-trends (red) and down-trends (blue) obtained from the segmentation of the mid-quote data, for up-trends (orange) and down-trends (green) obtained from the segmentation of the randomized mid-quote data with fixed zeros, and for up-trends (magenta) and down-trends (cyan) obtained from the segmentation of the totally randomized mid-quote data using patience levels: (a) $\tau = 60$ (1min); (a) $\tau = 600$ (10min); and (a) $\tau = 1800$ (30min). Insets show log-log plots.

The asymmetry between up- and down-trends is more explicit for the amplitudes: large down-trends (movements of depreciation of U.S. dollar against the Japanese yen), have higher probability than large trends in the opposite direction and can reach more extreme values of amplitude, e.g., $\sim 6$ JPY per USD in the $\tau = 1800$ (30min) case (Fig 10c). Such behavior is explained by the fact that the Japanese yen is seen as a safe-haven currency, a safe asset which protects investors during periods of uncertainty [32].

## Trend shape clustering

The study of the probability distributions above are important for the understanding of the market dynamics, but quantities such as length and amplitude summarize the whole trend in a single number and ignore its internal structure; we cannot know, for example, if a down-trend falls uniformly or if it accelerates. Aiming at a more detailed picture of USD/JPY market trends, we proceed to the investigation of trend shape, i.e., the relative position of all points (or a sample of points) of the trend.

Here we group similar trend shapes using cluster analysis so that we are able to describe the different types that occur in the USD/JPY mid-quote time series; we are particularly interested in finding trend shapes that are rare in the randomized data and possibly related to exceptional events. For such task we need to choose a measure of distance between trends that reflect their shapes and a clustering method. For the distance between trends, we normalize the trends by setting unit length and unit amplitude (that is, we rescale the original trend horizontally by its length and vertically by its absolute amplitude) and sample a fixed number of points from the normalized trend at fixed positions (we take 100 equidistant points); we define the distance between trends as the Euclidean distance between the vectors formed by the sampled points from the corresponding normalized trends. For example, if we have two perfectly linear trends, the distance between them is zero independent of their lengths or amplitudes, confirming that they have exactly the same shape; on the other hand, we can have trends with same length and amplitude but with distance greater than zero because they have distinct shapes. For the clustering method, we select the agglomerative hierarchical clustering with complete-linkage criterion: starting from clusters formed by individual trends, at each time step we merge the two clusters with the shortest distance, where the distance between clusters $X$ and $Y$ is defined as the maximum distance between a trend in $X$ and a trend in $Y$ [33–35].

We apply the described method to the trends obtained by the segmentation of the mid-quote time series of the currency pair USD/JPY from 2015 to 2018 using $\tau = 1800$ (30min), that is, focusing on macrotrends. We work with the subset of trends with absolute amplitude $|a| > 0.5$, filtering out small trends. In Fig 11 we present the dendrogram generated by the clustering process that shows the clusters of similar trend shapes and their relations. The first cluster in the left is the one containing all trends, which have as children clusters the cluster of all up-trends and all down-trends, which have their own children clusters until the last clusters in the far right corresponding to the individual trends. In the graphs, we show the normalized trends in the cluster by plotting all normalized trends in gray, the average in black and the standard deviation in pink. The clusters represented by red symbols are the ones whose trends have shapes that deviate from the randomized data case as explained next.

After grouping similar trend shapes, we look for the clusters deviating from the random case, i.e., the clusters containing trend shapes of rare occurrence in the randomized data. First, we take the randomized mid-quote data with fixed zeros and extract the trends using the segmentation with same patience level $\tau = 1800$ (30min) and condition $|a| > 0.5$. Next, for each cluster of the original data and each trend from the randomized data, we compute the distance between cluster and trend from randomized data (using the definition of distance between

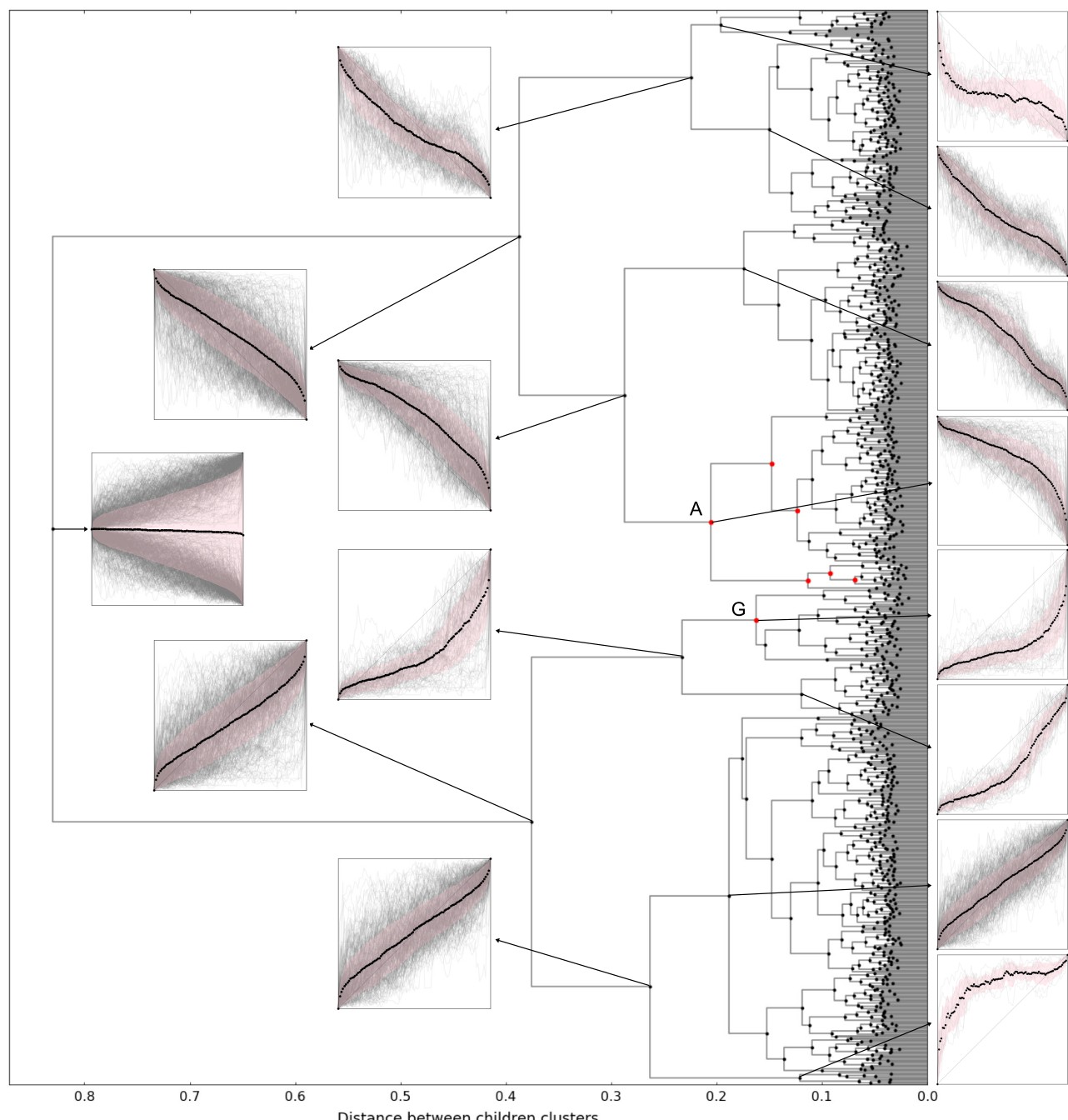

**Fig 11. Dendrogram indicating the similarities between shapes of trends obtained by the segmentation of the mid-quote time series of the currency pair USD/JPY from 2015 to 2018 using patience level $\tau = 1800$ (30min).** Only trends with absolute amplitude $|a|>0.5$ are considered. Each symbol represents a cluster of shapes and graphs show the normalized trends (gray lines), the average (black symbols) and the standard deviation (pink shade). Red symbols in the dendrogram indicate the clusters that deviate from the random case.

clusters) and count the number of such trends whose distance is shorter than the maximum distance between trends within the cluster. Finally, having for each cluster a number of trends from the original data and a number of trends with similar shapes from the randomized data, we apply the Fisher's exact test to check if the actual proportion of trends from original and

from randomized data in a cluster is incompatible with the proportion of the total trends of each category supposing the null hypothesis of randomly selecting trends to compose the cluster. The probability of grouping $n_{data}$ from a total of $N_{data}$ trends and $n_{rand}$ from a total of $N_{rand}$ trends assuming that all trends have the same probability of being chosen is given by the hypergeometric distribution [36]:

$$P(n_{data}, n_{rand}, N_{data}, N_{random}) = \frac{\binom{N_{data}}{n_{data}}\binom{N_{rand}}{n_{rand}}}{\binom{N_{data} + N_{rand}}{n_{data} + n_{rand}}}. \tag{11}$$

The total number of trends from the original data $N_{data}$ and from the randomized data $N_{rand}$ are fixed by the results of the segmentation: $N_{data}$ = 1055 trends and $N_{rand}$ = 1376 trends. The number of trends to be selected under the null hypothesis to compose each cluster is also fixed and equal to $n_{data} + n_{rand}$. We then use as p-value the probability of the number $n'_{data}$ of trends selected from $N_{data}$ trends under the null hypothesis being greater or equal to the observed $n_{data}$:

$$\text{p} - \text{value} = \sum_{n'_{data} \geq n_{data}} P(n'_{data}, n_{data} + n_{rand} - n'_{data}, N_{data}, N_{random}). \tag{12}$$

We apply the Fisher's exact test only to clusters where $n_{data} > n_{rand}$; the others are regarded as non-deviant. Table 1 lists the clusters for which p-value is below $10^{-5}$, interpreted as the clusters that deviate from the random case. The same clusters are shown in red in the dendrogram of Fig 11 and detailed in Fig 12. We remind that the deviations from the random case that we discovered are related solely to the trend shapes, disregarding trend length or amplitude.

For a more meticulous analysis, we turn our attention to cluster D, the largest one with no trend from the random case, i.e., no trend from the shuffled time series data has shape similar to the original USD/JPY market data trends in the cluster. The average trend shape of cluster D is characterized by a sharp fall in the end of the trend, with the last $\sim 10\%$ of length of the trend accounting for $\sim 80\%$ of its amplitude. Fig 13 depicts all 28 down-trends in cluster D with their original lengths and amplitudes and Table 2 details them (labels in the first column correspond to the ones in Fig 13): date of minimum (i.e., end of the trend), time of minimum, trend length, trend amplitude and associated event.

By searching for the date and time of the trends in specialized media, it was possible to identify associated events for 17 of the 28 trends, including 3 trends ((o), (p) and (q)) connected with the already mentioned Brexit Referendum in 2016 [37–39], highlighting trend (q) with extreme amplitude of $\sim 6$ JPY per USD when the victory of the Leave side was consolidating

**Table 1. Clusters of trend shapes that deviate from the random case.**

| Cluster | Trend type | $n_{data}$ | $n_{rand}$ | p-value |
|---------|-----------|-----------|-----------|---------|
| A | Down | 174 | 82 | $4.742 \times 10^{-17}$ |
| B | Down | 146 | 87 | $3.986 \times 10^{-10}$ |
| C | Down | 134 | 94 | $6.940 \times 10^{-7}$ |
| D | Down | 28 | 0 | $5.753 \times 10^{-11}$ |
| E | Down | 25 | 0 | $7.345 \times 10^{-10}$ |
| F | Down | 15 | 0 | $3.447 \times 10^{-6}$ |
| G | Up | 70 | 30 | $3.749 \times 10^{-8}$ |

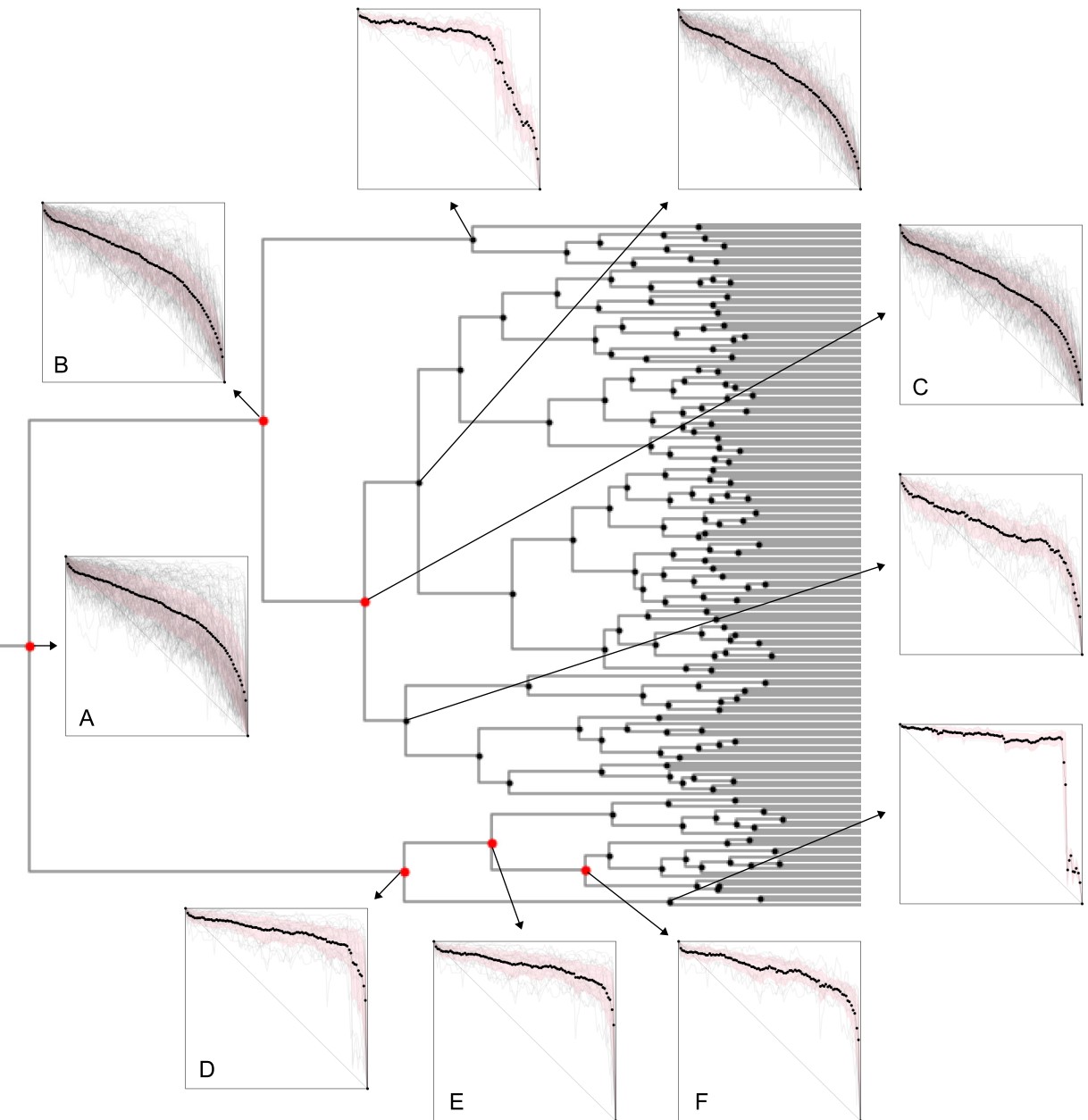

**Fig 12. Portion of dendrogram detailing the clusters of down-trend shapes that deviate from the random case.** Graphs show the normalized trends (gray lines), the average (black symbols) and the standard deviation (pink shade). Cluster labels correspond to the ones in Table 1.

(see down-trend distribution for mid-quote data in Fig 10c). Trends (c) and (d) correspond to the called China's Black Monday on 2015 August 24, when the Shanghai main share index fell 8.49% affecting other financial markets [40–42]. Trend (y) is linked with the 2016 United States elections won by Donald Trump [43, 44]. The remaining trends are related to monetary policy announcements from the central banking system of the United States, the Federal Reserve (Fed): trends (g) [45] and (z) [46, 47]; and from the Bank of Japan (BOJ): trends (f) [48], (h) [49, 50], (k) [51–53], (n) [54–56], (v) [57–59] and (x) [60–62]. In particular, the BOJ announcement on 2016 July 29 associated with trend (v) defined monetary easing actions to

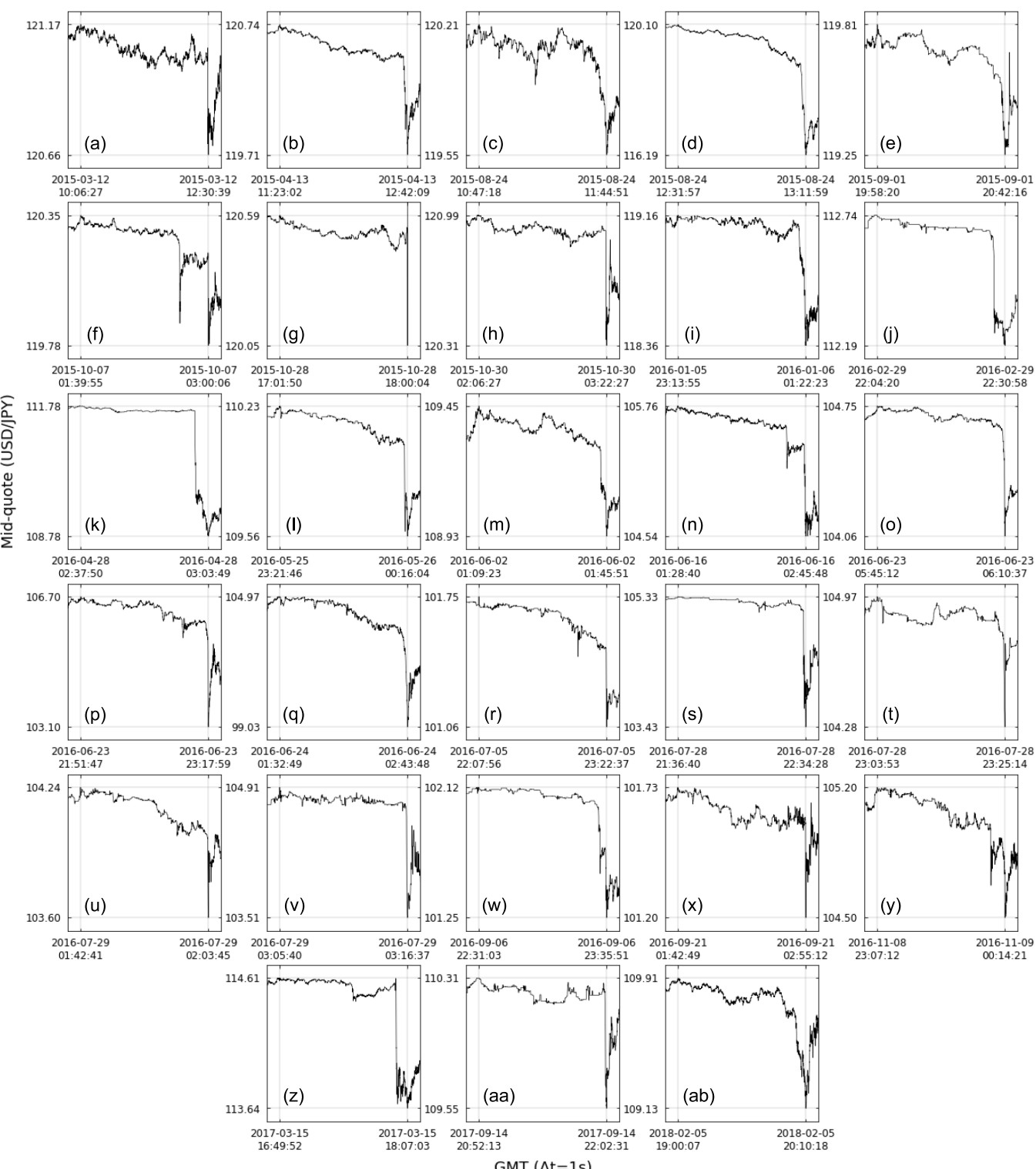

**Fig 13. All 28 down-trends of the USD/JPY market data from 2015 to 2018 in cluster D.** Shape of trends in this cluster are marked by a sharp fall in the end of the trend, having ∼80% of its amplitude in the last ∼10% of its length (trends are limited by the gray lines). See Table 2 for trends details.

**Table 2. Details of all 28 down-trends of the USD/JPY market data from 2015 to 2018 in cluster D.**

| Trend | Date of minimum | Time of minimum | Length | Amplitude | Associated event |
|---|---|---|---|---|---|
| (a) | 2015-03-12 | 12:30:39 | 8652 | -0.5050 | - |
| (b) | 2015-04-13 | 12:42:09 | 4747 | -1.0300 | - |
| (c) | 2015-08-24 | 11:44:51 | 3453 | -0.6650 | China's Black Monday |
| (d) | 2015-08-24 | 13:11:59 | 2402 | -3.9075 | China's Black Monday |
| (e) | 2015-09-01 | 20:42:16 | 2636 | -0.5625 | - |
| (f) | 2015-10-07 | 03:00:06 | 4811 | -0.5725 | BOJ Announcement |
| (g) | 2015-10-28 | 18:00:04 | 3494 | -0.5400 | Fed Announcement |
| (h) | 2015-10-30 | 03:22:27 | 4560 | -0.6775 | BOJ Announcement |
| (i) | 2016-01-06 | 01:22:23 | 7708 | -0.8025 | - |
| (j) | 2016-02-29 | 22:30:58 | 1598 | -0.5525 | - |
| (k) | 2016-04-28 | 03:03:49 | 1559 | -3.0100 | BOJ Announcement |
| (l) | 2016-05-26 | 00:16:04 | 3258 | -0.6800 | - |
| (m) | 2016-06-02 | 01:45:51 | 2188 | -0.5175 | - |
| (n) | 2016-06-16 | 02:45:48 | 4628 | -1.2175 | BOJ Announcement |
| (o) | 2016-06-23 | 06:10:37 | 1525 | -0.6850 | Brexit Referendum |
| (p) | 2016-06-23 | 23:17:59 | 5172 | -3.5975 | Brexit Referendum |
| (q) | 2016-06-24 | 02:43:48 | 4259 | -5.9425 | Brexit Referendum |
| (r) | 2016-07-05 | 23:22:37 | 4481 | -0.6900 | - |
| (s) | 2016-07-28 | 22:34:28 | 3468 | -1.9025 | BOJ Announcement[*] |
| (t) | 2016-07-28 | 23:25:14 | 1281 | -0.6825 | BOJ Announcement[*] |
| (u) | 2016-07-29 | 02:03:45 | 1264 | -0.6375 | BOJ Announcement[*] |
| (v) | 2016-07-29 | 03:16:37 | 657 | -1.4025 | BOJ Announcement |
| (w) | 2016-09-06 | 23:35:51 | 3888 | -0.8725 | - |
| (x) | 2016-09-21 | 02:55:12 | 4343 | -0.5250 | BOJ Announcement |
| (y) | 2016-11-09 | 00:14:21 | 4029 | -0.6975 | U.S. Election |
| (z) | 2017-03-15 | 18:07:03 | 4631 | -0.9700 | Fed Announcement |
| (aa) | 2017-09-14 | 22:02:31 | 4218 | -0.7650 | - |
| (ab) | 2018-02-05 | 20:10:18 | 4211 | -0.7775 | - |

Trend labels in the first column correspond to the ones in Fig 13.

[*] Those trends occurred hours before the BOJ Announcement associated with trend (v), but they are related to this event (see text).

stimulate investments (partially as a response to the Brexit Referendum result) that in fact disappointed investors, who were expecting more aggressive measures and caused strong speculation before the announcement itself, probably responsible for trends (s), (t) and (u) [63, 64]. We then have that trends in cluster D with associated events are either related to an exceptional event, causing the yen appreciation which supports its status as safe-haven currency, or the reaction of the market to central banks announcements. Note, however, that no associated events were found for the remaining 11 trends in cluster D and there are probably other major events associated with different trend shapes, reminding us that this is still an incipient study and that the relationship between trend shapes and market events needs to be further investigated.

## Final remarks

The epsilon-tau procedure proposed in this work extends previous methods to determine up- and down-trends in time series, particularly the epsilon-drawdown method; besides

considering a tolerance level to decide the end of a trend, it introduces a patience level, a kind of tolerance limit in the time axis that controls the time scales of trends, highlighting micro-trends if its value is small value and macrotrends if large.

We first studied the epsilon-tau procedure applied to discrete random walks. We derived exact expressions for marginal probability distributions of trend lengths and trend amplitudes, which, together with numerical results, confirmed the expected exponential decay when increments are independent. We explained how to use the epsilon-tau procedure to segment time series in alternating up- and down-trends by successively applying the method and the dependence of the segmentation result on the choice of the patience level value.

We then used the time series segmentation to analyze financial data represented by the USD/JPY mid-quote time series. The probability distributions of trend lengths and trend amplitudes for the market data were compared with the ones for randomized data. Specifically for amplitudes, the tails of the distributions for the market data are heavier than the ones for randomized data. We also observed an asymmetry between up- and down-trends: down-trends with large amplitude, corresponding to the appreciation of the JPY, happen more often than large up-trends and they can reach more extreme values. The status of safe-haven currency of the Japanese yen explains this asymmetry.

Finally, we carried out a more detailed analysis of the internal structure of the market macrotrends with the concept of trend shape. We grouped trends with similar shapes though the complete-linkage clustering and used the Fisher's exact test to identify clusters containing shapes that rarely occur in the random case. We found a particular cluster whose average trend shape is characterized by a sharp fall in the end of the trend, with no similar shape in the randomized data. For 17 of its 28 down-trends, we could associated the sharp mid-quote drops with exceptional events in the studied period—China's Black Monday in 2015, Brexit Referendum in 2016 and the 2016 U.S. elections—and with announcements from the Federal Reserve and Bank of Japan. This type of analysis shows the potential of using the epsilon-tau procedure for historical analysis of market trends, e.g., in which situations or what kind of events are responsible for trends with large amplitudes or uncommon shapes. Real-time market applications and uses in other fields remain for future works.

## Supporting information

**S1 Appendix. Trend length and trend amplitude marginal probability distributions from the epsilon-tau procedure for random walks.**
(PDF)

## Author Contributions

**Conceptualization:** Arthur Matsuo Yamashita Rios de Sousa, Hideki Takayasu, Misako Takayasu.

**Formal analysis:** Arthur Matsuo Yamashita Rios de Sousa.

**Funding acquisition:** Misako Takayasu.

**Investigation:** Arthur Matsuo Yamashita Rios de Sousa.

**Methodology:** Arthur Matsuo Yamashita Rios de Sousa, Hideki Takayasu.

**Project administration:** Hideki Takayasu, Misako Takayasu.

**Software:** Arthur Matsuo Yamashita Rios de Sousa.

**Supervision:** Hideki Takayasu, Misako Takayasu.

**Validation:** Arthur Matsuo Yamashita Rios de Sousa.

**Visualization:** Arthur Matsuo Yamashita Rios de Sousa.

**Writing – original draft:** Arthur Matsuo Yamashita Rios de Sousa.

**Writing – review & editing:** Arthur Matsuo Yamashita Rios de Sousa, Hideki Takayasu, Misako Takayasu.

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
