## [Decision Letter · Decision Letter 0]

29 Jul 2020

PONE-D-20-15593

Segmentation of time series in up- and down-trends using the epsilon-tau procedure

PLOS ONE

Dear Dr. Yamashita Rios de Sousa,

Thank you for submitting your manuscript to PLOS ONE. After careful consideration, I feel that it has merit but does not fully meet PLOS ONE’s publication criteria as it currently stands. Therefore, I invite you to submit a revised version of the manuscript that addresses the points raised during the review process.

Reviewers find this paper quite interesting and they suggest its publication after minor revision. However, one of them has major concerns that I share. I agree that the state of the art should consider some recently published papers. Apart from the suggested by the reviewers I would suggest also to revise https://doi.org/10.1371/journal.pone.0188814

My major concern (see comments of reviewer 1 and 2) is the limitation of this study to the case of the Brexit Referendum and the USDJPY currency pair. I would like to be sure that this is not an important limitation in the application of the results obtained in this paper.

However, I am sure that authors will be able to answer properly to all questions so my decision is minor revision.

We look forward to receiving your revised manuscript.

Kind regards,

J E. Trinidad Segovia

Academic Editor

PLOS ONE

Journal Requirements:

2. Please clarify in your Data availability statement whether others can obtain the same dataset.

We note that you have indicated that data from this study are available upon request. PLOS only allows data to be available upon request if there are legal or ethical restrictions on sharing data publicly. For more information on unacceptable data access restrictions, please see http://journals.plos.org/plosone/s/data-availability#loc-unacceptable-data-access-restrictions.

3. Please explain the rationale for the development of your tool in light of recent research in this area, clearly indicating which problem with existing tools you are addressing.

Please clearly report at the beginning of your methods or results section which were the key performance measures used to establish the validity and utility of your method. Please also report clearly which statistical analysis was used to establish robustness of performance measures.

Please note that PLOS ONE requires that experiments, statistics, and other analyses must be performed to a high technical standard and described in sufficient detail to allow for reproducibility of the study (http://journals.plos.org/plosone/s/criteria-for-publication#loc-3). To demonstrate the performance of the method, we would expect comparisons to be drawn between existing state-of-the-art methods.

4. Thank you for stating the following in the Financial Disclosure section:

"This study is supported by the Joint Collaborative Research Laboratory for MUFG AI Financial Market Analysis. The funder had no role in study design, data collection and analysis, decision to publish, or preparation of the manuscript."

We note that one or more of the authors are employed by a commercial company: Sony Computer Science Laboratories.

4.1. Please provide an amended Funding Statement declaring this commercial affiliation, as well as a statement regarding the Role of Funders in your study. If the funding organization did not play a role in the study design, data collection and analysis, decision to publish, or preparation of the manuscript and only provided financial support in the form of authors' salaries and/or research materials, please review your statements relating to the author contributions, and ensure you have specifically and accurately indicated the role(s) that these authors had in your study. You can update author roles in the Author Contributions section of the online submission form.

4.2. Please also provide an updated Competing Interests Statement declaring this commercial affiliation along with any other relevant declarations relating to employment, consultancy, patents, products in development, or marketed products, etc. 

Reviewers' comments:

Reviewer's Responses to Questions

**Comments to the Author**

1. Is the manuscript technically sound, and do the data support the conclusions?

Reviewer #1: Yes

Reviewer #2: Yes

Reviewer #3: Yes

2. Has the statistical analysis been performed appropriately and rigorously? 

Reviewer #1: Yes

Reviewer #2: Yes

Reviewer #3: Yes

3. Have the authors made all data underlying the findings in their manuscript fully available?

Reviewer #1: Yes

Reviewer #2: No

Reviewer #3: Yes

4. Is the manuscript presented in an intelligible fashion and written in standard English?

Reviewer #1: Yes

Reviewer #2: Yes

Reviewer #3: No

5. Review Comments to the Author

Reviewer #1: Overall and interesting and well written paper.

It is very very rare to see a paper that does not compare to some strawman. You need to compare to something, or make a strong case as to why you do not need to.

“we define the distance between trends as the Euclidean distance 207 between the corresponding normalized trends using the sampled points” Please expand, this is not clear to me.

Would two time series with identical slope have a zero distance?

Could you get similar results with a much similar way? For example, if you did PLA segmentation (your ref [2]) would that give you similar results to fig 4 etc

“For 17 of its 28 308 down-trends, we could associated the sharp mid-quote drops with exceptional events in 309 the studied period – China’s Black Monday in 2015, Brexit Referendum in 2016 and the 310 2016 U.S. elections – and with announcements from the Federal Reserve and Bank of 311 Japan” This strikes me as a little post-hoc

“The raw data used in this study was purchased from EBS Service Company Limited, with no special access privileges. Due to the contract between EBS and us, the authors are not allowed to distribute the raw data. Those researchers interested in analyzing similar data sets are recommended to contact with EBS Service Company Limited about the availability and purchase of the data (see https://www.cmegroup.com/tools-information/contacts-list/ebs-support.html).”

I will let this slide, but normally I do not accept papers until the code and data is all freely available.

Reviewer #2: In this paper the authors described a time series segmentation method called the epsilon-tau method. According to the authors, this method was first suggested by Johansen and Sornette in 2010 in passing but never fully developed. In this method, a time series segment no longer than a patience tau is called an uptrend if the ending value exceeds the starting value by greater than a tolerance epsilon. By varying epsilon and tau, and applying the method to the one-second time series data of the US dollar-Japanese yen exchange rate between 2015 and 2018, the authors examined a group of 28 highly significant downtrends and found that 17 of them can be attributed to known events.

This is an impressive piece of work. Overall, it is well written, and I recommend for it to accepted for publication, after the authors address my following minor concerns:

(1) I feel that the description of the epsilon-tau procedure needs to be more detailed, because what is written is somewhat confusing. For example, if a segment of the time series has x(t_end) - x(t_start) > epsilon, but at time t_mid, x(t_mid) - x(t_start) < epsilon, do we still consider this one segment (from t_start to t_end), or more than one segment (t_start to t_mid, and t_mid to t_end)?

(2) After understanding that the segmentation method is based on identifying robust local trends in the time series, I feel that it is similar in spirit to the DNA walk method by Peng et al. (Peng, C.K., Buldyrev, S.V., Goldberger, A.L., Havlin, S., Sciortino, F., Simons, M. and Stanley, H.E., 1992. Fractal landscape analysis of DNA walks. Physica A: Statistical Mechanics and its Applications, 191(1-4), pp.25-29), and should therefore cite this and similar papers.

Reviewer #3: The authors from the paper ‘segmentation of time series in up- and down-trends using the epsilon-tau procedure’ present a research work on time series in financial markets (in particular the currency exchange market) where time segmentation and trend analysis is of interest with regard to historical market data research. The document is correctly written, and the mathematical model is well presented, where I do not find major issues. Thus, the paper is ready for publication after revision, which involves:

1.- State of the art: At the introduction, the authors promptly jump into the topic of time series segmentation in the field of finance, namely financial markets. Here it would be good to introduce past and present research from the field. A good summary on market dynamics can be found at:

Joseph L. McCauley, Dynamics of Markets: The New Financial Economics. Cambridge University Press (2009)

Also, most recent research such as:

J. Clara-Rahola, A. M. Puertas, M. A. Sánchez-Granero, J. E. Trinidad-Segovia, and F. J. de las Nieves, Diffusive and Arrestedlike Dynamics in Currency Exchange Markets. Phys. Rev. Lett. 118, 068301 (2017).

Jan Jurczyk, Thorsten Rehberg, Alexander Eckrot, Ingo Morgenstern, Measuring Critical Transitions in Financial Markets. Scientific Reports 7, 11564 (2017).

Should be considered as these papers depict the time evolution on financial markets from a scope where dynamic periods are identified where markets, such as the currency exchange one, display physical phases as arrested crystal or glass states, clustered and random ones where prices diffuse. In these works, such states relate to phase transitions and risk management in a similar way in which the authors from the paper propose a time segmentation in market signals, in particular, the currency exchange one.

2.- Authors consider events for time series analysis. Here they focus on the Brexit Referendum from June 2016. This is a good period of analysis as short and middle-term fluctuations where significant in many markets due to the events in the UK. However, I wonder:

a. Why do authors focus only on the USDJPY currency pair? As they mention in the paper is the GBPUSD is the pair that displayed the largest fluctuations, decreasing down to historical magnitudes. Also, in this case, it would be worthwhile to consider the dynamics of other significant pairs such as the EURGBP or the GBPJPY. I don’t understand why the focus on the USDJPY only and will acknowledge I authors can explain this point.

b. Authors select the Brexit referendum and the USDJPY in their study. However, this is a single event. I will acknowledge if trend analysis and segmentation where discussed in terms of other events that influenced financial markets and in particular the Currency Exchange Market. For example, Trump’s victory on 2016, 09/11 and maybe most important to this study, the EURUSD correction in magnitude indirectly forced by the ECB with the Quantitative Easing purchase program. Here, the BCE directed a decrease of the EURUSD pair as the EUR was too high vs. the USD at the second half of the Great Recession, which helped in European exports and liquidity in the system. I encourage the authors to check for this data in the time period between late 2014 and late 2016 as a long-term negative trend is clearly observed. I think that adding such contents in the analysis, even if latter on the paper deeps in the USDJPY, would help in broadening the scope in which the proposed methodology can be considered and applied.

3.- Authors choose key magnitudes such as a patience level \\tau=7200 seconds from figure 7(a), or the one in figure 11 \\tau=1800 seconds. I might have missed it, but I do not fully understand why the selection of such magnitudes. What is particular to them?

4.- Probabilities and cumulative probabilities with trend amplitude or trend length clearly display an exponential character (as plotted in log-lin graphs), or similar. Here it would be helpful to quantify such data. It looks like that a stretched or expanded exponential could be a good fit (~exp(-(t/tc)^p), with tc a critical variable), where parameters such as p are significant of the type of fluctuations found in the currency and the time interval (for example random or correlated ones).

5.- Finally, it would be interesting to know about the scaling in the model. As exposed before, authors choose a particular currency and a particular time-frame. However, it would be interesting to know if the model scales if a broader or smaller sample is chosen (thus, another event). This is significant as literature states that events that induce strong volatility in different markets (such as Brexit, 9/11, etc) have a limited time extension after which markets stabilize. Thus, such fluctuations are high but improbable events, that disappear in long term due to market self-correcting. Even more, if scaling occurs, I would appreciate some comments related to Efficient Markets and the Efficient Market Hypothesis.

6. PLOS authors have the option to publish the peer review history of their article (what does this mean?). If published, this will include your full peer review and any attached files.

Reviewer #1: No

Reviewer #2: **Yes: **Siew Ann CHEONG

Reviewer #3: No

---

## [Author Response · Author response to Decision Letter 0]

13 Aug 2020

Please find the response to editor and reviewers in the file Response_to_Reviewers.docx

---

## [Decision Letter · Decision Letter 1]

28 Aug 2020

PONE-D-20-15593R1

Segmentation of time series in up- and down-trends using the epsilon-tau procedure

PLOS ONE

Dear Dr. Yamashita Rios de Sousa,

Thank you for submitting your manuscript to PLOS ONE. After careful consideration, I feel that it has merit but does not fully meet PLOS ONE’s publication criteria as it currently stands. Therefore, I invite you to submit a revised version of the manuscript that addresses the points raised during the review process.

Reviewers consider that most of the major concerns have been attended in this current version, however there are some minor issues that need to be addressed before this manuscript could be finally accepted.

We look forward to receiving your revised manuscript.

Kind regards,

J E. Trinidad Segovia

Academic Editor

PLOS ONE

Reviewers' comments:

Reviewer's Responses to Questions

**Comments to the Author**

1. If the authors have adequately addressed your comments raised in a previous round of review and you feel that this manuscript is now acceptable for publication, you may indicate that here to bypass the “Comments to the Author” section, enter your conflict of interest statement in the “Confidential to Editor” section, and submit your "Accept" recommendation.

Reviewer #1: All comments have been addressed

Reviewer #2: All comments have been addressed

Reviewer #3: All comments have been addressed

2. Is the manuscript technically sound, and do the data support the conclusions?

Reviewer #1: Yes

Reviewer #2: Yes

Reviewer #3: Yes

3. Has the statistical analysis been performed appropriately and rigorously? 

Reviewer #1: Yes

Reviewer #2: Yes

Reviewer #3: Yes

4. Have the authors made all data underlying the findings in their manuscript fully available?

Reviewer #1: Yes

Reviewer #2: No

Reviewer #3: Yes

5. Is the manuscript presented in an intelligible fashion and written in standard English?

Reviewer #1: Yes

Reviewer #2: Yes

Reviewer #3: Yes

6. Review Comments to the Author

Reviewer #1: I am happy with the changes

I am happy with the changes

I am happy with the changes

I am happy with the changes

Reviewer #2: I am satisfied with the revisions made by the authors in response to mine and other reviewers' comments.

Reviewer #3: The authors either applied appropriate modifications to the paper or provided valid explanations in reply to questions and concerns. However, there is one issue that should be considered. As stated in the previous review, the manuscript is focused on the USDJPY currency pair. Here, despite authors claim that \\epsilon - \\tau segmentation can be employed in other currencies (or markets). I wonder about the output in cumulative probability (either depending on trend length or amplitude) in the case of markets where volatility can be much remarkable such as the case of the EURUSD, or the opposite, low volatility markets such as the EURCHF.

In the case of the EURUSD and in the last one-two months (June-July 2020), the pair has displayed important trends as result of the health and economic impact of the covid19 disease at different scales. For example, the last Fed announcement was responsible for a highly volatile situation, with trends displaying significant amplitudes at high frequencies. However, at long times the pair displayed remarkable up-trends at least till about two to three weeks ago. I wonder about the probability and cumulative output in the case of the EURUSD, mainly due to its short-term volatility. For example, at trend lengths large enough, could cumulative probabilities (or probabilities) resolve echoes due to high frequency activity? Note that trends are characterized by channels, in practice defined by parallel or divergent Bollinger Bands. As well I wonder in the case of low volatility markets.

Therefore, and in lack of any analysis other than the one performed on the USDJPY, it would be clarifying to the contents of the manuscript that the title indicates that the study is performed on the USDJPY currency pair. Besides this change, I find that the last version submitted is of interest to PLoS ONE. If the authors consider mentioning the USDJPY pair in the title as focus of the study, the paper is ready for acceptance.

7. PLOS authors have the option to publish the peer review history of their article (what does this mean?). If published, this will include your full peer review and any attached files.

Reviewer #1: No

Reviewer #2: No

Reviewer #3: No

---

## [Author Response · Author response to Decision Letter 1]

28 Aug 2020

Please find our answers in 'Response_to_Reviewers.docx'.

---

## [Editor Report · Decision Letter 2]

8 Sep 2020

Segmentation of time series in up- and down-trends using the epsilon-tau procedure with application to USD/JPY foreign exchange market data

PONE-D-20-15593R2

Dear Dr. Yamashita Rios de Sousa,

I am pleased to inform you that your manuscript has been judged scientifically suitable for publication and will be formally accepted for publication once it meets all outstanding technical requirements.

Kind regards,

J E. Trinidad Segovia

Academic Editor

PLOS ONE
---

## [Editor Report · Acceptance letter]

10 Sep 2020

PONE-D-20-15593R2 

Segmentation of time series in up- and down-trends using the epsilon-tau procedure with application to USD/JPY foreign exchange market data 

Dear Dr. Yamashita Rios de Sousa:

I'm pleased to inform you that your manuscript has been deemed suitable for publication in PLOS ONE. Congratulations! Your manuscript is now with our production department. 

Kind regards, 

on behalf of

Dr. J E. Trinidad Segovia 

Academic Editor

PLOS ONE